# Molecular Marker-Assisted Selection for Frost Tolerance in a Diallel Population of Potato

**DOI:** 10.3390/cells12091226

**Published:** 2023-04-23

**Authors:** Wei Tu, Jingcai Li, Jianke Dong, Jianghai Wu, Haibo Wang, Yingtao Zuo, Xingkui Cai, Botao Song

**Affiliations:** 1National Key Laboratory for Germplasm Innovation & Utilization of Horticultural Crops, Key Laboratory of Potato Biology and Biotechnology, Ministry of Agriculture and Rural Affairs, College of Horticulture and Forestry Sciences, Huazhong Agricultural University, Wuhan 430070, China; 2Hubei Key Laboratory of Economic Forest Germplasm Improvement and Resources Comprehensive Utilization, Hubei Collaborative Innovation Center for the Characteristic Resources Exploitation of Dabie Mountains, College of Biology and Agricultural Resources, Huanggang Normal University, Huanggang 438000, China; 3College of Biological and Food Engineering, Hubei Minzu University, Enshi 445000, China

**Keywords:** frost tolerance, potato, diallel population, BSA, MAS

## Abstract

A multi-parental population is an innovative tool for mapping large numbers of loci and genetic modifications, particularly where they have been used for breeding and pre-breeding in crops. Frost injury is an environmental stress factor that greatly affects the growth, development, production efficiency, and geographical distribution of crops. No reported study has focused on genetic mapping and molecular marker development using diallel populations of potatoes. In this study, 23 successful cross combinations, obtained by a half diallel cross among 16 parents, including eight frost-tolerant advanced breeding lines and eight cultivars, were used to map the genetic loci for frost tolerance and to create a molecular marker-assisted selection (MAS) system. Three candidate regions related to frost tolerance on chromosomes II, V, and IX were mapped by bulked segregant analysis (BSA). Furthermore, six SNP markers associated with frost tolerance from candidate regions were developed and validated. Above all, a MAS system for the frost tolerance screening of early breeding offspring was established. This study highlights the practical advantages of applying diallel populations to broaden and improve frost-tolerant germplasm resources.

## 1. Introduction

Frost injury is one of the major environmental stress factors which drastically influences the growth, development, productivity, and geographic distribution of crop plants [1,2]. Potato (*Solanum tuberosum* L.), the third largest crop in terms of human consumption after rice and wheat, plays a crucial role in maintaining global food security and sustainable agri-food systems, notably in developing countries with high levels of poverty, hunger, and malnutrition [3]. Unfortunately, potato cultivars do not have frost tolerance or the genetic variability to develop it, so they suffer from serious freezing injuries when the temperature falls below −3 °C for even just a few hours [4,5]. Unlike disease and pest problems in field production, cultural and chemical treatments seem unlikely to help guard against frost damage; thus, it is imperative to create new potato cultivars with frost tolerance via genetic improvement.

Despite the lack of frost-tolerant germplasm in potato cultivars, several wild species, including *S. acaule*, *S. albicans*, *S. commersonii*, *S. malmeanum*, and *S. demissum*, have been recognized to possess strong frost tolerance and considerable interspecies or intraspecies genetic variation [5,6,7]. The outstanding frost-tolerant wild species, *S*. *commersonii* (2x, 1EBN, endosperm balance number) and *S. acaule* (4x, 2EBN), can tolerate an acute freezing episode up to −4.5 °C and −6 °C before cold acclimation and withstand freezing to as low as −11.5 °C and −9 °C after cold acclimation for several days [6]. Thus, the potentially useful freezing-hardiness characteristic derived from *S*. *commersonii* and *S. acaule* has been widely used to genetically improve the frost tolerance of cultivated potatoes [8,9,10,11]. However, the direct crossing of the two species with tetraploid *S. tuberosum* cultivars has sometimes been hampered with because of postzygotic barriers resulting from the parental genome imbalance in the endosperm [9]. To circumvent sexual isolation and incorporate frost tolerance into cultivated potatoes, protoplast fusion is most commonly employed between *S. commersonii* and *S. tuberosum*. The obtained frost-tolerant somatic hybrids can be successfully employed in frost-tolerance breeding by hybridizing with potato cultivars [12,13,14]. In addition to protoplast fusion, the introgression of frost tolerance derived from *S. commersonii* into cultivated species has also been performed by sexual means through the manipulation of ploidy and 2n gamete selection [8,9]. Although some breeding materials with improved frost tolerance have been obtained, the partially recessive frost tolerance inheritance of *S. commersonii* has to be considered during the process of the genetic improvement of cultivars.

On the other hand, a cross incompatibility resulting from the imbalance of EBN hampered the introgression of frost tolerance genes from *S. acaule* into tetraploid cultivated potatoes; even so, some breeding efforts with *S. acaule* and the further use of this species for promoting frost tolerance have been attempted via interspecific hybridization with the help of ploidy breeding and somatic fusion [8,15,16,17]. What is more interesting is that a frost-tolerant potato cultivar called Alaska Frostless, which combines the frost tolerance of *S. acaule* with desirable characteristics of potato cultivars and can tolerate field frosting as low as −3 °C for 2 h, was selected in 1961 at Matanuska [18]. Alaska Frostless has a significantly lower total yield and slower vine growth than common potato cultivars, resulting in large-scale cultivation and application being limited despite its culinary characteristics being impeccable. Although some progress has been made in the introduction of frost tolerance to wild potatoes into cultivated potatoes, there is still a lack of frost-tolerant potato cultivars that are suitable for large-scale cultivation and popularization.

The capacity of different plants to withstand freezing tolerance under natural conditions varies among plant species and cultivars [19,20,21]. The genetic studies of various plants have found that freezing (frost) tolerance is a complex polygenic trait, that is, controlled by a few loci that are responsible for most genetic variability [22,23]. The two major genetic components of freezing tolerance, non-acclimated freezing tolerance (NA, normal growing condition) and cold acclimation capacity (ACC, increase in freezing tolerance in response to chilling temperature), have been demonstrated to be independent genetic controls with a partial recessive inheritance [24]. Additionally, few independent genes are involved in the genetic control of NA and ACC according to a genetic analysis of frost tolerance in three segregating populations, namely a cross population and two backcross populations between freezing-tolerant (*S. commersonii*) and freezing-sensitive wild species (*S*. *cardiophyllum*) [24,25]. Notably, NA and ACC vary greatly between potato species [6]. In addition, two quantitative trait loci (QTLs) for NA, together with another two QTLs for ACC, were detected in separate genomic regions that were preliminarily narrowed down to a part of chromosome V due to a relatively small progeny size [26]. Recently, four QTLs related to frost tolerance were detected via BSA technology and traditional QTL mapping on Chr02 and Chr11 under natural low-temperature frost [27]. To date, there has been no follow-up research on *S. commersonii* genetic control in frost (freezing) tolerance and a molecular marker-assisted selection system for frost-tolerant potato breeding is still blank.

In this study, 23 cross combinations obtained by a half-diallel cross among 16 parents were used to create potato frost-tolerant resources and to establish a molecular marker-assisted system. Three candidate regions on chromosomes II, V, and IX were mapped by BSA. SNP markers were developed and validated for the candidate regions. In addition, six molecular markers were identified that were significantly correlated with frost tolerance and enabled the establishment of a MAS system for the early frost tolerance screening of breeding offspring. This study highlights the practical advantages of applying diallel populations to broaden and improve potato frost-tolerant genetic resources.

## 2. Materials and Methods

### 2.1. Plant Materials and Crossing Experiment

The experimental materials comprised eight interspecific accessions and eight potato cultivars. The eight interspecific accessions, with a pedigree of *S. commersonii*, *S. acaule*, and *S. tuberosum*, were 14FT04-25, 14FT04-44, 14FT04-63, 14FT04-71, 14FT24-10, 14FT43-25, 14FT51-08, and 14FT51-03 (Appendix A). These interspecific accessions were obtained from the Inter-Regional Potato Introduction Station at Sturgeon Bay, Wisconsin (https://npgsweb.ars-grin.gov/gringlobal/cooperator?id=53268, accessed on 20 April 2023), which were yielded by ploidy manipulation and bridge species and demonstrated a strong freezing tolerance with unexpected agronomic traits [8]. The eight potato cultivars with freezing susceptibility but excellent agronomic characteristics were Bora Valley, Pentland Crown, M1, M3, Plain, Denali, RH89-039-16, and Hua cai 1 (Appendix A), of which RH89-039-16 and Huacai 1 were from the Key Laboratory of Potato Biology and Biotechnology of Huazhong Agricultural University, and the others were from the Inter-Regional Potato Introduction Station at Sturgeon Bay, Wisconsin. All plants were propagated in vitro on an MS medium supplemented with 4% sucrose and 0.8% agar at 22 ± 2 °C with a photoperiod of 16 h/day under a light intensity of 60 µmol m^−2^ s^−1^.

The crossing experiments of the 16 parents (eight interspecific materials and eight cultivars) were designed for half-diallel crossing as described by Wu and Matheson [28], which included all the possible crosses between the parents involved in the cross in one direction, with each parent having an equal opportunity to mate and recombine with every other parent. The four-week-old, cultured seedlings of each crossing parent were planted into plastic pots (32 cm diameter) in a greenhouse where the conditions were favorable for potato growth and flowering. Twenty plants of each parent were planted twice for the crossing experiment, in which 16 plants were employed to emasculate and the others to collect the pollen for follow-up crossing. The 16 diverse parents were crossed by in a 16 × 16 half-diallel fashion [28]. The time of flowering in all crossing parents was approximately the same. The cross procedure was performed as described by Bamberg et al. [8]. All the crosses were carried out on intact plants in a greenhouse (HZAU, Wuhan). A total of 136 crosses were conducted, followed by half-diallel crossing, but only 23 cross combinations yielded viable seeds. The obtained viable seeds were sterilized and treated with 1.5 mg·mL^−1^ of gibberellin for 24 h to break dormancy, washed twice with distilled water when the seeds germinated, and finally transferred into a potato breeding nursery for tuber propagation. The 23 frost-tolerant crosses mentioned above were started in 2016 and so were named 16FT01, 16FT02, 16FT05, etc. (Appendix A).

### 2.2. Determination of Frost Tolerance

The parents and F_1_ hybrids were evaluated under a natural field frost experiment following a standard protocol [5]. The eight interspecific accessions and eight potato cultivars were planted at Wuhan (30.5° N, 114.4° E) on 21 October 2016, and Luoyang (34.6° N, 112.5° E) on 14 October 2016. To best visually identify the frost tolerance of parents and to select a suitable low temperature for the frost test of the F_1_ hybrids, long-term field trials concerning the parents were carried out on 21 January 2017 (Environment I), 11 February 2017 (Environment II), 9 January 2018 (Environment III), and 11 January 2018 (Environment IV) at Wuhan and on 15 December 2016 (Environment V), 27 December 2016 (Environment VI), and 28 December 2016 (Environment VII) at Luoyang and were assessed according to the local weather forecast. Moreover, a temperature recorder (RC-4HC), located 5 cm above the ground, was used to monitor the actual field temperature, and sufficiently low temperatures (below 0 °C) were recorded to indicate frost temperature (Appendix A). Combining the lowest temperature recorded with the frost performance of multiple parents across the seven environments, Environments I and IV seemed to effectively characterize various degrees of frost tolerance in the F_1_ hybrids (Appendix A).

The F_1_ hybrids mentioned above were grown on 21 October 2016, and on 28 September 2017, and the frost tolerance was evaluated in Environment I and Environment IV. The F_1_ hybrids and cross parents, with five replicates for each genotype per environmental condition, were planted into plastic pots (32 cm diameter) at random spots in a polytunnel greenhouse where interplant spacing was approximately 30–40 cm. After approximately two months of growth for these materials under normal conditions, the canopy of the polytunnel greenhouse was removed under appropriate frost temperature conditions that combined local historical temperature records with a local real-time weather forecast prediction system. The standard and extent of the frost damage were scored visually on a 0–6 scale according to the methods of Vega and Bamberg [5]. Each plot was then carefully assessed and scored when the frosts were severe enough to differentiate the hardiness of standard species, and the average injury score (AS) was represented as frost tolerance. AS = (X0*N0 + X1*N1+ X2*N2 + X3*N3 + X4*N4 + X5*N5 + X6*N6)/(N0 + N1 + N2 + N3 + N4 + N5 + N6), where X0~X6 is the scale value of 0 to 6 according to Vega and Bamberg [5] for each individual, and N0~N6 represents the numbers of the respective scales for each individual. Variation within accessions was evaluated by the response of each of the five individual plants.

### 2.3. Genomic DNA Extraction, Bulking, and Sequencing

The total genomic DNA of parents and F_1_ hybrids was isolated and purified from 3-week-old young leaves using the Hi-DNA Secure Plant Kit DP350 (TIANGEN, Beijing, China) for subsequent sequencing. Then, DNA concentration was quantified using a Nanodrop 1000 spectrophotometer (Thermo Scientific, Waltham, MA, USA). The RP (resistant pool collected from frost-tolerant individuals) and SP (sensitive pool collected from sensitive-tolerant individuals) were constructed by mixing an equal amount of the total DNA for each individual. Each pool included 30 individuals exhibiting not only extreme phenotypes for frost tolerance but also the progenies of frost-tolerant parents that were selected in equal proportions in RP or SP (Appendix A; Table 1). DNA libraries were constructed using the NEBNext Ultra II DNA Library Prep Kit for Illumina (New England Biolabs, MA, USA), and the high-throughput sequencing of DNA libraries was carried out by the Illumina NovaSeq platform with a NovaSeq 6000 S4 Reagent Kit in Wuhan Genoseq Technology Co., Ltd., Wuhan, China. 

### 2.4. Identification of the Genomic Region Responsible for Frost Tolerance

To obtain high-quality data, fastp and Trimmomatic were first used to clean the sequence reads and filter out poor-quality bases (<Q10), adapter sequences, and primers [29,30]. Subsequently, the data were mapped to the *S. tuberosum* group Phureja DM reference genome (v6.1) using the Burrows–Wheeler Aligner (BWA) tool (BWA version: 0.7.5a-r405, Wellcome Trust Sanger Institute, Cambridgeshire, UK) [31,32], and then the sequence alignment map (SAM) format alignment results were obtained. SAMtools v 0.1.19 (Wellcome Trust Sanger Institute, Cambridgeshire, UK) was used to convert the SAM alignment results to BAM files and remove the potential PCR duplicates [33]; these reads were then sorted into BAM files using SortSam in Picard tools v1.91 (http://broadinstitute.github.io/picard/, accessed on 20 April 2023). Processed alignment files (.bam) were created for each pool and were used for subsequent variant calling analyses. Finally, variation detection (including Indels and SNPs) was performed using the “HaplotypeCaller” module of GATK software v3.7 (The Broad Institute of Harvard and MIT, Cambridge, MA, USA) [34]. ED (Euclidean distance) was calculated as described by Hill et al. [35]. The CMplot package was used to draw the density distribution of SNPs and Indels along each chromosome.

### 2.5. Snp Marker Development via Genotyping-by-Sequencing (GBS)

Based on the BSA-seq interval located above, target region capture sequencing was employed to develop the SNP marker for frost tolerance. In total, 80 SNP target sites, 40 from chromosome V and the other from chromosome IX (Appendix A) were selected based on the value of the SNP index and Δ(SNP-index), belonging to the located region of frost tolerance. SNP target sites whose SNP-index between RP and SP exceeded 0.25 were selected and then mapped to the potato DM genome to obtain a DNA sequence of 150–200 bp around the SNP site. The specific PCR primers were designed based on the DM reference genome (v6.1) by the NCBI (National Center for Biotechnology Information) online primer design tool Prime-blast (https://www.ncbi.nlm.nih.gov/tools/primer-blast/, accessed 20 April 2023). These primers were synthesized by Sangon Biotech Co., Ltd. (Shanghai, China).

A total of 80 specific primers were first used to capture the DNA sequences of the parents and the progenies among extreme bulks in each target site, which then recovered the PCR products that passed quality control to construct the paired-end (PE) sequencing library. These libraries were then sequenced on the Illumina HiSeq 3000 platform, which generated 150 bp paired-end reads (GenoSeq, Wuhan, China). For the raw data, Fastqc software (http://www.bioinformatics.babraham.ac.uk/projects/fastqc/, accessed on 20 April 2023) was first used to check the quality of the raw data, and fastp and Trimmomatic were used for data quality control to obtain high-quality clean data [29,30]. BWA software (BWA version: 0.7.5a-r405, Wellcome Trust Sanger Institute, Cambridgeshire, UK) was employed to assign the clean data to each target site and obtain the mapping SAM format file [32], followed by SAM format conversion to the BAM format through SAMtools v 0.1.19 (Wellcome Trust Sanger Institute, Cambridgeshire, UK) [33]. “SortSam” in “Picard” tools v1.91 (http://broadinstitute.github.io/picard/, accessed 20 April 2023) was used to sort the reads in the BAM document and then removed the PCR repeats to obtain the final BAM file. Finally, the “HaplotypeCaller” module of GATK software v3.7 (The Broad Institute of Harvard and MIT, Cambridge, MA, USA) was used to ascertain the genotype of each target locus of the parents and the progenies between extreme bulks.

### 2.6. Statistical Analysis

The SNP markers that expressed different genotypes at the same loci between frost-tolerant and frost-sensitive parents were considered to be polymorphic and were selected for GBS. The DNA samples of the parents and all progenies in each population were analyzed by the selected polymorphic markers as described above. The result of genotyping by GBS was scored for each marker used for the assay as described above, and the polymorphic markers were selected for each population. Each SNP allele between frost-tolerant and frost-sensitive parents that was significantly correlated with frost tolerance was scored as ‘1’ or ‘0’, respectively. Bivariate correlation analysis was used to assess the relationship between the frost tolerance and genotypic data of the progenies, and the Spearman correlation coefficient was used to estimate the significance of this correlation. A marker was considered to be closely associated with a gene when the marker was significant at *p* < 0.05, with a correlation coefficient of *p* > 0.3 between genotypic and phenotypic data [36].

## 3. Results

### 3.1. Evaluation of Frost Tolerance of Parents and Progenies

Frost injury ratings were scored to evaluate the frost tolerance under field conditions of eight interspecific hybrids and eight potato cultivars. Two field frost trials concerning the cross parents were performed under a low temperature based on the local weather forecast station at Wuhan (Environments I, II, III, IV) and Luoyang (Environments V, VI, VII) (Appendix A). The frost tolerance of cross parents in Environments I, III, and VI showed that the eight cultured potatoes expressed some frost damage after the field of natural frost (Figure 1a,b), and the average AS of the eight cultured potatoes was 3.49 ± 0.68 (Environment I), 3.80 ± 0.78 (Environment III) and 5.18 ± 0.80 (Environment VI), respectively (Figure 2a,b; Appendix A). However, the eight interspecific hybrids showed extremely strong frost tolerance with hardly any frost injury under the same environments as above (Figure 1a,b), with average AS values of 0.19 ± 0.24, 0.58 ± 0.52, and 0.10 ± 0.20, respectively (Figure 2a,b; Appendix A). For the other environments, the frost in Environments II, IV, and VII froze all of the cultured potatoes to death as a result of the extended time that the plants were subjected to freezing temperatures (Appendix A). Some interspecific hybrids exhibited stronger frost tolerance, and the average AS of the eight interspecific hybrids was 2.15 ± 2.12, 2.20 ± 1.24, and 2.39 ± 0.98 in Environments II, IV, and VII (Figure 2a,b; Appendix A). The frost tolerance of frost-tolerant and frost-sensitive parents in the various environments showed that there were significant differences in the average frost tolerance regardless of the environment (Figure 2a,b). The eight interspecific hybrids exhibited stronger frost tolerance, indicating that they could provide a material basis for the frost-tolerance breeding of cultivated species.

A total of 406 progenies were obtained from 23 cross combinations derived from 16 parents, including eight frost-tolerant interspecific hybrids and eight cultivars, referring to the half-diallel crossing design. On the basis of the frost tolerance of multiple parents under different field experiments, the frost test in Luoyang either directly froze all the cultivated potatoes to death or caused hardly any damage (Appendix A; Figure 2a,b; Appendix A), which is not suitable for differentiating between various grades of resistant breeding materials. In contrast, the appropriate frost time and temperature in Wuhan enabled the effective characterization of various degrees of frost tolerance for potato germplasm materials (Appendix A, Figure 2a,b). Given the lowest temperature recorded by a recorder along with the frost performance of multiple parents across the seven environments (Appendix A), Environments I and IV seemed to effectively characterize various degrees of frost tolerance in F_1_ hybrids. Therefore, these sprouted and robust tubers of cross parents and progenies were grown in a polytunnel greenhouse to secure uniform conditions in the autumn of Sep 2016 and Oct 2017. After removing the canopy of the polytunnel greenhouse and allowing complete thawing, there were obvious visual differences in the frost performance between parents and progenies in Environments I and IV (Figure 1). The frost performance was scored by the evaluation system of Vega and Bamberg [5].

AS was employed to express the frost tolerance of each genotype under different environments. The average AS of the 406 progenies was 2.22 ± 1.22 (Environment I) and 2.74 ± 1.42 (Environment IV), and the coefficients of variation (CV) were 54.93% and 51.71%, respectively. The frost tolerance of the 406 progenies under two different environments exhibited a continuous distribution (Figure 2c,d), indicating how to frost tolerance is a quantitative trait controlled by multiple genes. In addition, the correlation analysis of frost tolerance showed that there was a highly significant correlation between the two frost tests (2.22 vs. 2.74; *p* < 0.01, two-tailed *t*-test). The average frost tolerance of all progenies was significantly different from that of the frost-tolerant and frost-sensitive parents. The average frost tolerance of the progenies was significantly stronger than that of the frost-sensitive parents but significantly lower than that of the frost-tolerant parents (Appendix A).

### 3.2. Identification of the Genomic Region Responsible for Frost Tolerance

Bulk segregation analysis (BSA) was carried out to identify the genomic region that was responsible for frost tolerance under natural frost conditions. To ensure that the progenies of frost-tolerant parents were selected in equal proportions from the extreme pool of the diallel population, 30 individuals with extreme frost tolerance and 30 individuals with extreme frost sensitivity were selected from 23 cross combinations to construct the RP (resistant pool) and SP (sensitive pool), respectively. The frequency distribution of the frost-tolerant parents between RP and SP showed that the average frequency (4.13 ± 1.89 vs. 4.00 ± 2.20) was almost consistent with the total frequency (33 vs. 32) of the frost-tolerant parents (Table 1), indicating that the genome dose of the frost-resistant parents between RP and SP was relatively balanced. An analysis of frost tolerance between RP and SP in Environments I and Environment IV showed that the frost tolerance of RP was significantly higher than that of SP under these two environmental conditions (Figure 3).

After the sequencing conducted on the Illumina HiSeq sequencing platform, 90,170,687 and 102,494,884 reads were produced from RP and SP, with mean Q20 and Q30 values of 97.42% and 92.65%, respectively (Appendix A). These sequences were aligned to the reference genome DM_V6_1 using BWA/SAM software with an average alignment rate of 82.98% and 85.28% between RP and SP. The average genome coverage and depth were 75.96% and 23.47×, respectively (Appendix A).

The statistical methods of ED (Euclidean distance) were employed to identify the candidate regions that were associated with frost tolerance. A total of 622,836 Indels and 6,457,501 SNPs distributed on 12 chromosomes were obtained. The distribution of all SNPs across the whole genome is shown in Appendix A. The ED value of each identified SNP with observed and expected allele depths were calculated, and the value trends were plotted to show the distribution across chromosomes 1–12 in a 1 Mb sliding window with a step size of 100 kb (Figure 4). Combining the information of ED along with chromosomes 1–12, three candidate regions, 39.19–42.70 Mb located on chromosome II, 22.34–44.80 Mb in chromosome V, and 12.17–22.51 Mb, 32.25–49.15 Mb located on chromosome IX, were significantly correlated with frost tolerance (*p* < 0.01). Hence, the three candidate regions of chromosome II, chromosome V, and chromosome IX located by MMAPPR (Mutation Mapping Analysis Pipeline for Pooled RNA-seq) were considered to be responsible for frost tolerance, suggesting that multiple genetic loci may be responsible for frost tolerance.

### 3.3. Development of SNP Markers for Frost Tolerance

In the present study, three candidate regions related to frost tolerance were mapped (Figure 4), while the marker development for two candidate regions of chromosomes V and IX was accomplished due to the fine mapping of the candidate region of chromosome II, which was currently progressing. The SNPs with a Δ index of more than 0.25 between RP and SP were selected in candidate regions as described above. Forty SNPs located on chromosome V and 40 SNPs on chromosome IX were uniformly screened in the corresponding candidate regions to develop frost-tolerant SNP markers and target capture sequencing was performed for SNP polymorphic site detection and genotyping between frost-tolerant and frost-sensitive parents. After genotyping, the SNP type for the frost-tolerant parents was assigned a value of ‘1’, and the sensitive parent was assigned a value of ‘0’ (Table 2), e.g., the SNP type of the marker Chr05V42 was divided into two types among cross parents (G/T and G/G, G/T only distributed in frost-tolerant parents and referred to as 1), while only the G/G type (referred to as 0) was distributed in sensitive parents at the same loci (Appendix A). The results of parental SNP genotyping showed that only 67 SNP markers were detected at least once between the eight frost-tolerant and eight frost-sensitive parents, accounting for 83.75%, of which a total of 42 SNP markers exhibited SNP polymorphisms, which accounted for 62.69% (Appendix A). The correlation of parental genotypic data from 42 polymorphic SNP sites with frost tolerance revealed that 22 SNP markers had a significant correlation in different environments (Appendix A). Subsequently, SNP genotyping was performed on each individual from the two extreme bulks using the 22 markers described above, and the correlation and goodness-of-fit analysis of the genotypic data with frost tolerance in different environments demonstrated that only nine SNP markers were significantly and negatively correlated with frost tolerance in Environments I and IV (Table 2). Except for the marker chr09V97 in Environment I (*p* < 0.05), the markers showed a highly significant correlation in each environment (*p* < 0.01), and the mean goodness of fit between the nine SNP markers and the frost tolerance phenotype was 73.89%, ranging from 65.52% to 82.00% (Table 2).

To further verify the performance of the nine markers, 330 progenies from 23 cross combinations were genotyped, and correlation analysis between the genotypic data and frost tolerance was performed. The results showed that Chr05V42, Chr05V92, Chr05V158, Chr09V212, Chr09V222, and Chr09V26 were significantly correlated with frost tolerance (*p* < 0.01) (Table 3).

### 3.4. MAS for Frost Tolerance

To clarify the effect of each SNP allele individually (‘1’ or ‘0’) that was associated with the frost tolerance of all progenies, the frost tolerance of each SNP allele genotyped ‘1’ or ‘0’ was analyzed. The results show that the frost tolerance conferred by the ‘1’ allele was significantly stronger than that conferred by the ‘0’ allele in both Environment I and IV (Appendix A), suggesting that frost tolerance could be controlled by multiple genetic loci.

To further investigate the different marker combinations derived from chromosome V and chromosome IX that was responsible for frost tolerance, each progeny was genotyped through the use of the above six markers (marker order chr05V42, chr05V92, chr05V158, chr09V26, chr09V212, and chr09V222). After genotyping, the genotypes of progenies were distinguished as four types, namely, chr05(0) + chr09(0) (six marker genotypes of chromosomes V and IX were overall ‘0’), chr05(0) + chr09(1) (three marker genotypes of chromosome V were ‘0’ overall, at least one marker of chromosome IX was ‘1’, and what follows was similar), chr05(1) + chr09(0), and chr05(1) + chr09(1), comprising 21, 52, 8, and 245 genotypes, respectively. The frost tolerance of different marker combinations suggested that chr05(1) + chr09(1) exhibited the strongest frost tolerance, and there were significant differences in comparison with the other three types in Environment I or IV (Figure 5).

Since chr05(1) + chr09(1) exhibited the best frost tolerance and had the most progenies, it was essential to categorize the genotypes of this marker combination in detail. A total of 12 types of chr05(1) + chr09(1) genotypes were identified (Table 4), of which the ‘111111’ type had the most progenies, accounting for 40.74% (99/243). An assessment of the frost tolerance of each type of genotype showed that there was no significant difference among the various genotypes in Environment I, while frost tolerance varied somewhat between the different genotypes in Environment IV, with the ‘001011’ type demonstrating the worst frost tolerance and the ‘111111’ type demonstrating the strongest frost tolerance. There was no significant difference between the other types of frost tolerance. Combining the Spearman correlation with the goodness of fit for these six SNP markers, chr05V158, chr09V26, and chr09V212 appeared to have better frost tolerance on the condition that these SNP genotypes were ‘1’. In addition, when chr05V158, chr09V26, and chr09V212 were all ‘1’, such as in genotypes ‘001111’, ‘011111’, ‘101111’, and ‘111111’, the plant exhibited a stronger frost tolerance than the others, especially in Environment IV (Table 4). Conversely, the frost tolerance appeared to be poor if chr05V158, chr09V26, or chr09V212 lacked at least one ‘1’ allele, e.g., ‘001011’, ‘100011’, and ‘100111’. When combined with the correlation and coincidence between SNP markers and genotype and frost tolerance phenotypes, the SNP marker combination chr05V158+ chr09V26+ chr09V212, with chr05V158 of chromosome V together with chr09V26 and chr09V212 located on chromosome IX, could effectively screen for frost-tolerant progeny.

## 4. Discussion

A typical frost-tolerant wild *Solanum* species confers superior frost tolerance to cultivated species, and much research has been conducted on the generation of frost-tolerant germplasm derived from *S. commersonii* and *S. acaule* [5,6,12,13,37]. However, there is an extreme lack of information concerning MAS breeding for frost tolerance, especially in the wild species *S. acaule* and *S. commersonii*. In this study, a diallel population was used to incorporate frost-tolerant germplasm into cultivars and to develop frost-tolerant SNP markers via target region capture sequencing. Finally, a MAS system was established for frost tolerance screening in breeding offspring.

Freezing tolerance is a complex genetic trait that is controlled by multiple genes [22], but much less is known about the mechanisms underlying the differences in freezing (frost) stress tolerance among species. As an outstanding wild species with strong freezing tolerance, the inheritance of freezing resistance in *S. commersonii* was proposed [24], but little research has focused on the genetic mapping of freezing tolerance. Compared with other methods of determining freezing tolerance [38,39,40], the frost assay using natural frost in the field, in which plants are directly exposed to a natural subzero temperature environment, appears to more authentically assess frost tolerance than other detection methods [27]. Bulked segregant analysis (BSA) can be used to find molecular markers associated with a trait by detecting the variation present in bulks of segregants that have been sorted per phenotype [36]. Moreover, the simplicity and low cost of BSA have allowed it to be widely applied to decipher complex traits, including those whose genetic control is unknown, and it has been increasingly used to detect quantitative trait loci (QTL) [41]. In the present research, eight interspecific accessions with the pedigree of *S. commersonii*, *S. acaule*, and *S. tuberosum*, which were yielded by ploidy manipulation and bridge species [8], were identified as having strong frost tolerance in various natural frost environments (Appendix A; Figure 1). Hence, it can be speculated that *S. acaule* and *S. commersonii* may react to frost tolerance simultaneously. The diallel population obtained by the half-diallel cross of 16 parents was used to locate genetic loci for frost tolerance, and three candidate regions located on chromosomes II, V, and IX were mapped using BSA (Figure 4). Vega et al. [26] mapped two QTLs concerning a non-acclimated freezing tolerance and two other cold-acclimation genetic loci, both on chromosome V, through the *S. commersonii* BC_1_ population. In addition, the candidate regions for frost tolerance on chromosome II were detected in a BC_1_ segregated population that was constructed by the frost-tolerant wild species *S. commersonii* with the frost-sensitive wild species *S. verrucosum* [27]. Combining the previous genetic mapping of frost (freezing) tolerance with candidate regions in our study, the results show that frost resistance could be regulated by different loci under variant low-temperature pressures, which is consistent with the results of rice [42,43]. Thus, the results of genetic mapping imply that the genetic loci of frost tolerance located on chromosomes II and V could come from *S. commersonii*, while the genetic loci of chromosome IX might be attributed to the wild species *S. acaule*. Consequently, mapping these findings could provide an important reference for the fine mapping of frost tolerance and the cloning of frost-tolerant genes.

Molecular marker-assisted breeding schemes can greatly improve the accuracy and efficiency of breeding regardless of the growth stage and environment. In *Solanum* species, molecular markers that are resistant to PVY, *Verticillium* wilt, and late blight have been developed and applied to the MAS breeding scheme [44,45,46]. To date, hardly any molecular markers associated with frost tolerance have been found by genetic mapping in potatoes, let alone by the MAS program for freezing tolerance. Li [47] developed the CAPS marker linked with the *SAD* gene in a BC_1_ population, which was derived from the diploid frost-tolerant wild species *S. commersonii* crossed with frost-sensitive *S. cardiophyllum*. In the present study, a half-diallel population was developed by a half-diallel cross among 16 parents, including eight frost-tolerant advanced breeding lines and eight cultivars. Theoretically, this should have generated 136 cross-populations, but 23 cross-combinations were actually produced because of unilateral incompatibility or self-incompatibility. After 2 years of frost tests, extreme frost-tolerant, and frost-sensitive bulks were used for the genetic mapping of frost tolerance using BSA, and three candidate regions were ultimately located on chromosomes II, V, and IX (Figure 4). Moreover, target region capture sequencing was carried out to successfully develop six SNP markers and to design a molecular marker-assisted selection system associated with frost tolerance in the half-diallel population. Because of the complexity of the tetraploid potato genome, the development of frost-tolerant molecular markers is mainly based on the diploid level, and there are currently no molecular markers that are related to the tetraploid level. In the present study, we developed the first six SNP markers that were significantly correlated with frost tolerance at the tetraploid genome level by using target region capture sequencing; three SNPs were located on chromosome V, and three were on chromosome IX (Table 3). To develop a molecular marker-assisted selection for frost tolerance, each progeny was genotyped using six screened SNP markers to investigate the different SNP marker combinations that were responsible for frost tolerance. Combining the Spearman correlation with the goodness of fit for these six SNP markers, it was found that the SNP markers chr05V158, chr09V26, and chr09V212 appeared to have better frost tolerance when these SNP alleles were ‘1’. In addition, genotypes such as ‘001111’, ‘011111’, ‘101111’, and ‘111111’ exhibited stronger frost tolerance than the others, especially in Environment IV, since in these genotypes, the markers chr05V158, chr09V26, and chr09V212 all had the ‘1’ allele (Table 4). Hence, the SNP chr05V158 of chromosome V combined with chr09V26 and chr09V212 located on chromosome IX could enable the effective screening of frost-tolerant progeny (chr05V158 + chr09V26 + chr09V212). With the development of modern genomics, the cost of sequencing decreased year by year, so the MAS strategy for large numbers of offspring became a more efficient and comprehensive breeding scheme that could greatly accelerate the frost tolerance breeding process.

Over the last decade, multi-parental populations have become a preferred focus of genetics research in various crop species due to an increased probability that a polymorphic allele could be present in one subpopulation and provide higher allelic diversity for QTL mapping than bi-parental populations [48]. The diallel mating design is most commonly used in maize to analyze the general (specific) combining ability and heterosis for the traits of grain yield, grain quality traits [49,50,51], high oil content [52], and abiotic stresses [53]. In addition, the analysis of multi-parental populations has been demonstrated to be an interesting strategy for identifying the sources of allelic variation, which are directly useful for MAS and for detecting epistatic interactions in a diallel cross between four unrelated maize lines [54]. However, to our knowledge, no study has focused on the genetic mapping or MAS of frost tolerance underlying a diallel population in potatoes. Genotyping-by-sequencing (GBS) can rapidly identify and genotype many SNPs, and GBS was used in the present study to genotype a diallel population, which now enables the subsequent identification of high-resolution markers associated with a given trait. In addition, the half diallel crossing program for mapping quantitative traits provides a new mapping resource for the detailed genetic dissection of complex agronomic traits on an elite genetic background breeding scheme in potatoes [55], and QTL mapping together with additive effect estimation for frost tolerance is ongoing by the researchers involved in the present study.

In conclusion, the present research succeeded in locating genetic loci for frost tolerance on chromosomes II, V, and IX through a diallel population. Frost-tolerant SNP markers were also developed using target region capture sequencing, and eventually, a MAS system was created at the tetraploid level for frost tolerance screening. Due to its high efficiency and low cost, the MAS system could greatly accelerate the process of frost-tolerant breeding.

## Figures and Tables

**Figure 1 cells-12-01226-f001:**
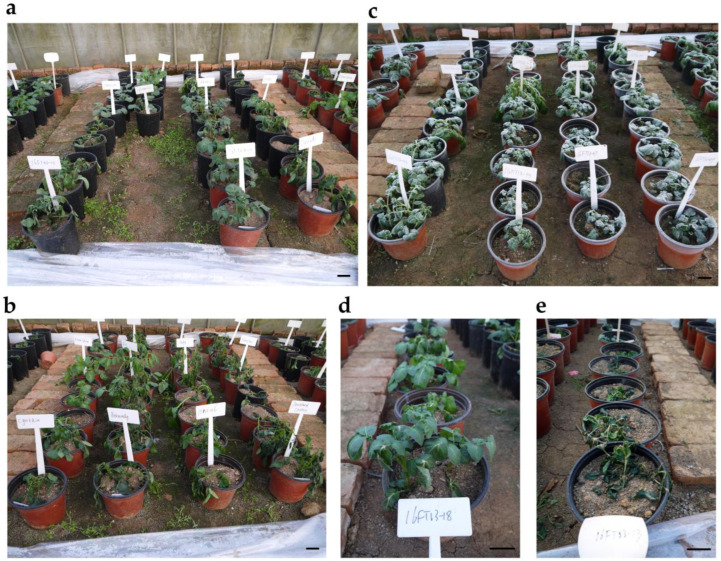
The performance of parents and their hybrids in natural frost conditions. The performance of some frost-tolerant parents (**a**) and frost-sensitive parents (**b**) after frost in Environment I. The canopy of the polytunnel greenhouse was removed at p.m. on 10 February 2018, under a local real-time weather forecast prediction system. The F_1_ hybrids were subjected to natural frost in the field (**c**). Frost performance was scored at p.m. on 11 February 2018 (Environment IV) after plants were thoroughly thawed. The performance of resistance genotypes after meeting frost in Environment IV (**d**), and the performance of sensitive genotypes after meeting frost in Environment IV (**e**). Bars = 5 cm.

**Figure 2 cells-12-01226-f002:**
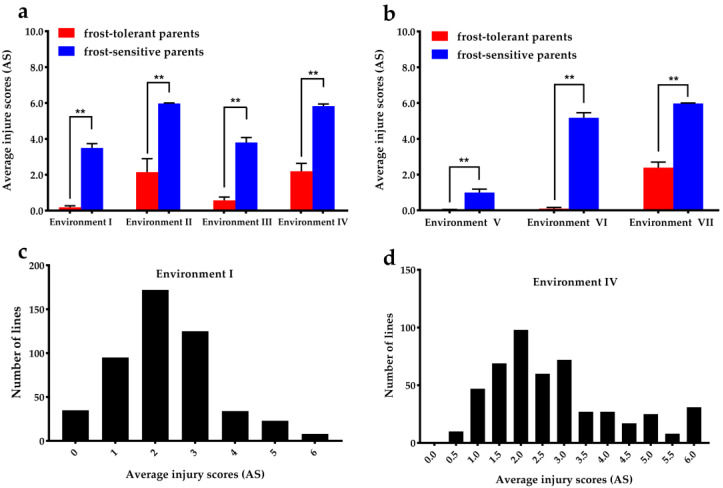
The comparison analysis and frequency distribution of frost tolerance of parents and progenies in various environments. The crossing parent’s frost performance, eight frost-tolerant interspecific hybrids and eight frost-sensitive cultivated potatoes, were evaluated in four environments (Environment I, II, III, IV) at Wuhan (**a**) and three environments (Environment V, VI, VII) at Luoyang (**b**), respectively. Frequency distribution of the average injury scores (AS) for frost tolerance of the 23 cross combinations in Environments I (**c**) and IV (**d**). Values are the mean of at least three biological replicates. Bars indicate SD. Asterisks indicate significant differences using Student’s *t*-test (** *p* < 0.01).

**Figure 3 cells-12-01226-f003:**
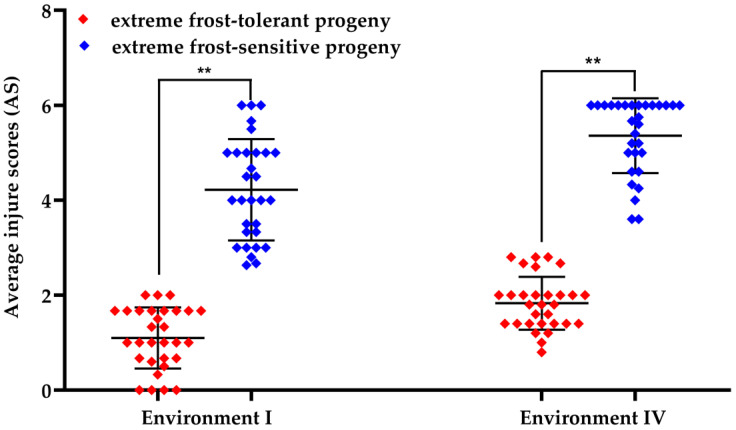
Comparison with frost tolerance of extreme pools in Environments I and IV. Asterisks indicate significant differences using Student’s *t*-test (** *p* < 0.01).

**Figure 4 cells-12-01226-f004:**
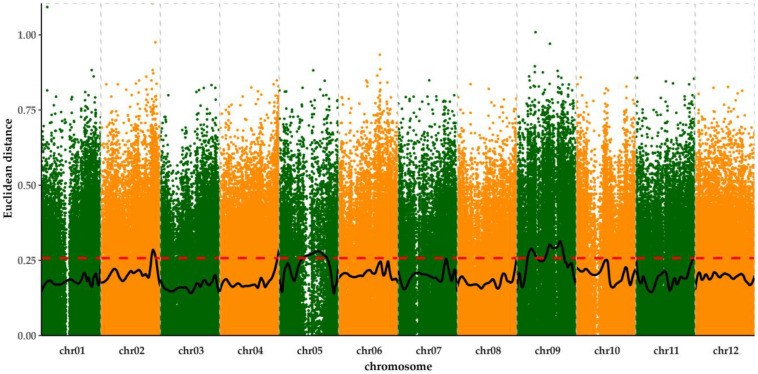
Mapping of genetic loci for frost tolerance. The *X*-axis represents the position of the SNPs on 12 chromosomes and the *Y*-axis represents the association value of the SNPs calculated by ED (Euclidean distance). The ED distribution was visualized by a 1 Mb sliding window method with a step size of 100 kb at about 90% confidence along all the 1–12 chromosomes. The horizontal red dotted line is the default threshold, which is three standard deviations above the genome-wide median, with a value of 0.277 in this figure.

**Figure 5 cells-12-01226-f005:**
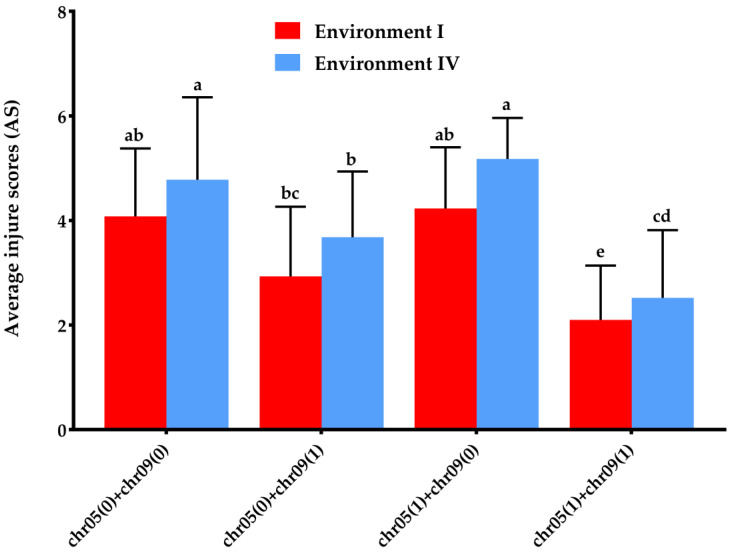
Comparative analysis of frost tolerance of different types of marker combinations from chromosomes V and IX. Values are the means ± SD (n ≥ 3). Lower cases indicate significant differences compared to each other using Tukey’s multiple comparisons tests (*p* < 0.05).

**Table 1 cells-12-01226-t001:** Frequency distribution of frost-tolerant (frost-sensitive) parents between the extremely sensitive pool and resistant pool.

Frost-Tolerant Parents	Resistant Pool	Sensitive Pool	Frost-Sensitive Parents	Resistant Pool	Sensitive Pool
14FT43-25	3	4	RH89-039-16	4	4
14FT04-25	3	6	Bora Valley	0	2
14FT04-44	5	6	Pentland Crown	3	0
14FT04-63	3	2	M1	3	0
14FT04-71	5	3	Hua cai 1	0	2
14FT24-10	8	5	M3	5	2
14FT51-08	4	6	Plain	0	1
14FT51-03	2	0	Denali	7	19
Total	33	32	total	22	30
Average	4.13 ± 1.89	4.00 ± 2.20	average	2.75 ± 2.60	3.75 ± 6.30

**Table 2 cells-12-01226-t002:** The nine SNP markers in relation to frost tolerance in extreme pools.

SNP Name	SNP Genotypes	No. in Resistant Pool	No. in Sensitive Pool	Extreme Pools		
				Spearman Correlation		Goodness of Fit (%)
				Environment I	Environment IV	
chr05V42	G/T (1)	24	12	−0.559 **	−0.447 **	73.21
	G/G (0)	3	17			
chr05V81	A/C (1)	24	9	−0.624 **	−0.575 **	77.78
	A/A (0)	3	18			
chr05V92	A/A (1)	14	2	−0.559 **	−0.440 **	75.93
	A/G (1)	11	9			
	G/G (0)	2	16			
chr05V158	G/G (1)	22	6	−0.583 **	−0.517 **	82.00
	A/G (0)	2	18			
	A/A (0)	1	1			
chr09V26	G/G (1)	10	0	−0.490 **	−0.561 **	76.79
	A/G (1)	11	7			
	A/A (0)	6	22			
chr09V59	T/T (1)	12	2	−0.368 **	−0.483 **	67.24
	A/T (0)	17	27			
chr09V97	T/T (1)	9	0	−0.292 *	−0.417 **	72.41
	C/T (1)	17	13			
	C/C (0)	3	16			
chr09V212	G/G (1)	10	0	−0.624 **	−0.559 **	74.14
	A/G (1)	18	14			
	A/A (0)	1	15			
chr09V222	A/A (1)	13	4	−0.441 **	−0.515 **	65.52
	A/C (1)	16	16			
	C/C (0)	0	9			

*, ** Significant at the 0.05 and 0.01 probability level, respectively; brackets inside ‘0’ and ‘1’ indicate the frost-tolerant and -sensitive SNP type.

**Table 3 cells-12-01226-t003:** The SNP markers related to frost tolerance in 330 progenies.

SNP Name	SNP Genotypes	Spearman Correlation		Chromosome
		Environment I	Environment IV	
chr05V42	G/T (1)	−0.315 **	−0.403 **	V
	G/G (0)			
chr05V92	A/C (1)	−0.361 **	−0.433 **	V
	A/G (1) A/A (0)			
chr05V158	G/G (1)	−0.316 **	−0.363 **	V
	A/G (0)			
	A/A (0)			
chr09V26	G/G (1)	−0.317 **	−0.489 **	IX
	A/G (1)			
	A/A (0)			
chr09V212	G/G (1)	−0.458 **	−0.432 **	IX
	A/G (1)			
	A/A (0)			
chr09V222	A/A (1)	−0.369 **	−0.378 **	IX
	A/C (1)			
	C/C (0)			

** Significant at the 0.01 probability level; brackets inside ‘0’ and ‘1’ indicate the frost-tolerant and -sensitive SNP type.

**Table 4 cells-12-01226-t004:** The SNP genotyping and frost tolerance analysis of each genotyping type.

Type of SNP Genotype	Number of SNP Genotypes	Environment I		Environment IV	
		Frost Tolerance of Progeny	Multiple Comparisons (*p* < 0.05)	Frost Tolerance of Progeny	Multiple Comparisons (*p* < 0.05)
001011	11	2.73 ± 1.09	ns	4.39 ± 1.37	a
100111	3	3.50 ± 1.32	ns	3.73 ± 2.02	ab
100011	4	2.75 ± 1.27	ns	3.60 ± 1.62	ab
110011	18	2.39 ± 0.90	ns	3.14 ± 1.63	ab
101011	15	1.99 ± 0.85	ns	2.94 ± 1.21	b
111011	29	2.12 ± 1.13	ns	2.80 ± 1.30	b
011011	9	2.35 ± 1.09	ns	2.73 ± 1.20	bc
001111	14	1.98 ± 0.80	ns	2.62 ± 1.17	bc
110111	16	2.36 ± 0.52	ns	2.53 ± 1.37	bc
011111	16	2.15 ± 0.91	ns	2.23 ± 0.92	bc
101111	9	1.85 ± 1.28	ns	2.04 ± 0.87	bc
111111	99	1.84 ± 1.02	ns	2.00 ± 0.92	c

Lower cases indicate significant differences compared to each other using Tukey’s multiple comparisons tests (*p* < 0.05). ‘ns’ indicates no significant differences.

## Data Availability

All data generated or analyzed in the current study are included in the published article and the additional data are provided as Appendix A.

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
