# Peer review of "Molecular Marker-Assisted Selection for Frost Tolerance in a Diallel Population of Potato"

_cells, 2023, doi:10.3390/cells12091226_

Round 1
Reviewer 1 Report
Minor corrections are required. Corrections are marked in text.
There is need to rewrite the discussion in light of suggestions.

Author Response
Author's Reply to the Review Report (Reviewer 1)
Comments and Suggestions for Authors
Minor corrections are required. Corrections are marked in text. There is need to rewrite the discussion in light of suggestions.
Response: We greatly appreciate your review of our manuscript and your constructive comments. In the following part, we made point-to-point responses to each comment. As the reviewer suggested, we have rewritten the discussion to clearly deliver the message.
- Line 32: “Frost injury, one of the major environmental stress factors, drastically .....“ need reorganization
Response: Thanks for your valuable advice. We have reorganized it in revision (line 33-34) according to the reviewer’s comments.
- Line 38: should delete the word “...will...”.
Response: Thanks for your suggestion. We have deleted it as the reviewer's advice in revision (line 39).
- Line 110: delete phrase “... in this research...”
Response: Thanks for the advice. We have corrected it as the reviewer's suggestion in revision (line 125).
- Line 115-116: “...and bridge species and exhibited strong freezing tolerance but unexpected agronomic traits...” should replace the word “exhibited” as “have” and replace the word “but” as “with”
Response: Thanks. We have corrected it as the reviewer's comments in the revision (line 132-133).
- Line 123: should delete the word “.....”
Response: Thanks for your suggestion. We have deleted it as the reviewer's suggestion (line 141).
- Line 132: “Flowering in all crossing parents came about approximately at the same time....” should rewrite.
Response: We sincerely appreciate such a professional and valuable issue. We have corrected it as “ Time of flowering in all crossing parents was approximately same” in line 150-152.
- Line 133-136: “The flowering parents were timely emasculated and the emasculated stigmas were then wrapped in a butter paper....”: no need of detail. Cite standard reference for method of crossing.
Response: We are very grateful for your constructive comments. We have revised the manuscript to make it more concise as follows: The cross procedure was performed as described by Bamberg et al (lines 152).
- Line 145-146, “The evaluation of the frost tolerance of parents and F1 hybrids was performed with a natural field frost trial following a published protocol [5]...” should rewrite.
Response: We sincerely appreciate such a professional issue. We have corrected it as “The parents and F1 hybrids were evaluated under natural field frost experiment following a standard protocol” in line 165-167.
- Line 149, Line 242, Line 261: “...multiparent ...” should replace as “...”
Response: Thanks for your suggestion. We have corrected it throughout the revision according to the reviewer's suggestion (line 150, line 241, line 254).
- Line 151. “... January 9, 2018 (Environment III), and January 11, 2018 (Environment IV) ...” only two day difference between sowing dates, it is not significant difference between two days to scan them different environments.
Response: We apologize for the lack of clarity in this part of the manuscript. In the present study, two field frost trials concerning the cross parents were firstly performed under low temperature based on the local weather forecast station at Wuhan (Environments I, II, III, IV) and Luoyang (Environments V, VI, VII) (Fig S1). As shown in Fig S1, the lowest temperature on January 9, 2018 was -2.1℃, while the lowest temperature on January 11, 2018 decreased to -5.5℃ with snowing (Fig S1). The frost tolerance of parents in Environments III showed that the eight cultured potatoes expressed some frost damage after the field natural frost while the eight interspecific lines showed hardly any frost injury in the field (Fig 2a; Table S3). The frost in Environments IV froze the whole cultured potatoes to death as a result of a long time at freezing temperatures while some interspecific hybrids exhibited stronger frost tolerance (Fig 1a, b; Fig 2a; Table S3).
- Line 250, Line 255: “line” should replace “hybrids”
Response: Thanks for your suggestion. We have modified it consistently throughout the revision as the reviewer's suggestion (line 280, line 286).
- Line 266, “Figure 1. The frost performance of frost tolerance of multiparent and progenies in natural frost” need reorganization.
Response: Thanks for your valuable advice. We have re-created Figure 1 along with Figure lends in revision as the reviewer's advice (line 299-306).
- Line 311: should delete the word “...various...”
Response: Thanks for your suggestion. We have deleted it as the reviewer's suggestion (line 356).
- Line 375: table 3 “No. of sensitive pool” should correct “No. in sensitive pool”
Response: Thanks for your comments. We have corrected it as the reviewer's suggestion (line 428).
- Line 383: the word “...with...” should correct “in”
Response: Thanks for your advice. We have corrected it in line 436 as the reviewer's suggestion.
- Line 482, line 512: “In this ..” should correct “In present study...”
Response: Thanks for your comments. We have corrected it in line 511 and line 540 as the reviewer's suggestion.
- line 512-521 the part of “In this ..half diallel population” should move to between line 479 and line 480
Response: We sincerely appreciate the reviewer for bringing up such good advice. We will take your suggestions and move this section to the previous paragraph in line 540-550.
- Line 521-523 the sentence “A named diaQTL...three parents ” should delete
Response: Thank you for taking the time to review our manuscript and pointing out the issue with the reference. We have deleted it as the reviewer's comments (line 594).
Reviewer 2 Report
How authors fulfilled the assumptions of diallel analysis since, it is asexually propagated crop?
There is no novelty in this article
Pronouns are frequently use in the article, avoid this in revised version
There is several errors, like
However, the postzygotic barriers resulting from the imbalance in endosperm balance 52 number (EBN) of the endosperm have sometimes hampered the direct crossing of the two species with tetraploid S. tuberosum cultivars despite their encompassing a rich source of frost tolerance genes [9]. (This is very poor text, and not understandable for readers……
circumvent their sexual isolation and incorporate frost tolerance into the cultivated potato, protoplast fusion is most commonly employed to obtain 56 frost-tolerant somatic hybrids between the wild species S. commersonii and S. tuberosum, and successful interspecific hybridization between these somatic hybrids and potato cultivars has also been carried out to further improve the characters of these hybrids (This is another example that one sentence consist of almost four lines, poor text.
What are the basis of selection of wild species use in material methods? What is genetic origin, any previous studies.
This study was carried out in 2018, how authors give validity of results?
Many errors must be replaced with subscript/superscript with F (filial generation)
This article must be rephrased with the help of native English speaker and it should not be accepted in current form for publication.
Author Response
Author's Reply to the Review Report (Reviewer 2)
Comments and Suggestions for Authors
- How authors fulfilled the assumptions of diallel analysis since, it is asexually propagated crop?
Response: Thank you for your thorough review and for raising this important point. The vegetative (tuber) planting material of clonally propagated potatoes is typically called seed potato, which is widely used in the production of commercial potatoes. The term true potato seed (TPS) was introduced for tetraploid potato varieties produced from botanical seed, which is used for genetic improvement and the development of new varieties. As potato is a cross-fertilizing crop developed from heterozygous parents that do not ‘breed true’, TPS is heterogeneous and thus variable for a large number of characteristics. All the genetic advantages of clonally propagated crops can be used for variety development, and the genotype that will finally be released is among the progeny immediately after the initial crossing.
The diallel mating design was most commonly used in maize for genetic variation analysis. As for potato, Killick (1977) analyzed the general combining ability effects that were responsible for differences in maturity by a half-diallel set of crosses with six parents for the first time. In recent years, the diallel mating design has been used to study genetic diversity and evaluate genetic resources (Darabad et al., 2020; Ruiz de Arcaute et al., 2022; Vesali et al., 2020). And a named diaQTL software for QTL mapping in autotetraploid diallel populations was developed, and the software applied to analyze tuber shape in a tetraploid potato diallel with three parents (Amadeu et al., 2021).
- Amadeu RR, Muñoz PR, Zheng C, Endelman JB. QTL mapping in outbred tetraploid (and diploid) diallel populations. Genetics, 2021, 219.
- Darabad GR, Hassandokht MR, Hassanpanah D, Mousavi A. Diallel cross in potato cultivars (Solanum tuberosum) and evaluation of their progenies under deficit water stress. Acta Agrobotanica, 2020, 73: 1-9.
- Ruiz de Arcaute R, Carrasco A, Ortega F, Rodriguez-Quijano M, Carrillo JM. (2022/05/09, 2022.) Evaluation of Genetic Resources in a Potato Breeding Program for Chip Quality. Agronomy 10.3390/agronomy12051142.
- Vesali MR, Baradaran R, Hassanpanah D, Seghatoleslami MJJRdAN. Generating genetic diversity through diallel crosses of promising potato cultivars (Solanum tuberosum L.) and studying cultivar hybrids under water deficit stress. Revista de Agricultura Neotropical, 2020, 7: 49-56.
- There is no novelty in this article
Response: We sincerely appreciate the reviewer for bringing up the issue. Genetic studies of various plants have found that freezing (frost) tolerance is a complex polygenic trait that is controlled by a few loci. The potato is a highly heterozygous autotetraploid, until now, few findings have been reported on the genetic mapping of potato frost resistance, let alone a molecular marker-assisted selection system for frost-tolerant potato breeding under field conditions. The diallel mating design has been used to study genetic diversity and evaluate genetic resources, however, hardly any in genetic mapping and molecular marker development in potato.
In the present study, three candidate regions of frost tolerance were mapped by a half diallel mating design among 16 parents, of which the genetic loci locating chromosome II is a new finding, and the fine-mapping of the narrower candidate region of chromosome II currently being finely mapped in progress. In addition, we obtained six molecular markers significantly correlated with frost tolerance and established the marker-assisted selection system for early frost tolerance screening of breeding offspring. Our study highlights the practical advantages of applying diallel populations to broaden and improve potato frost-tolerant genetic resources.
- Pronouns are frequently use in the article, avoid this in revised version.
Response: We apologize for any confusion caused by the pronouns incorrect use in the previous manuscript. With the help of Elsevier's language service, we have carefully checked and improved the English writing including the use of pronouns in the revised manuscript accordingly.
- There is several errors, like
However, the postzygotic barriers resulting from the imbalance in endosperm balance number (EBN) of the endosperm have sometimes hampered the direct crossing of the two species with tetraploid S. tuberosum cultivars despite their encompassing a rich source of frost tolerance genes [9]. (This is very poor text, and not understandable for readers……
circumvent their sexual isolation and incorporate frost tolerance into the cultivated potato, protoplast fusion is most commonly employed to obtain 56 frost-tolerant somatic hybrids between the wild species S. commersonii and S. tuberosum, and successful interspecific hybridization between these somatic hybrids and potato cultivars has also been carried out to further improve the characters of these hybrids (This is another example that one sentence consist of almost four lines, poor text.
Response: Thank you for your valuable comments. We have corrected it as the reviewer's comments in revised manuscript: “However, the direct crossing of the two species with tetraploid S. tuberosum cultivars has sometimes been hampered because of the postzygotic barriers resulting from the parental genome imbalance in the endosperm. To circumvent sexual isolation and incorporate frost tolerance into cultivated potato, protoplast fusion is most commonly employed between S. commersonii and S. tuberosum. The obtained frost-tolerant somatic hybrids were successfully employed in frost-tolerance breeding by hybridizing with potato cultivars” (line 55-67).
- What are the basis of selection of wild species use in material methods? What is genetic origin, any previous studies.
Response: Thank you for your thorough review and for raising this important point. Firstly, the superior frost-tolerant wild species, S. commersonii (2x, 1EBN) and S. acaule (4x, 2EBN) can tolerate an acute freezing episode to -4.5 °C and -6 °C before cold acclimation and withstand freezing as low as -11.5 °C and -9 °C after cold acclimation for several days (Chen et al., 1980) (line 48-51). A cross incompatibility resulting from the imbalance in EBN hampered S. acaule and S. commersonii cross directly with tetraploid cultivated potatoes. Bamberg et al (1994) used chromosome-doubled S. commersonii (4x, 2EBN) as a bridge species to cross with S. acaule, resulting in fertile F1 interspecific hybrids. Some of these F1 hybrids produced 2n gametes, which enabled direct crossing to S. tuberosum, resulting in 6x interspecific hybrids (Bamberg et al., 1994). These 6x interspecific hybrids with a pedigree of S. commersonii, S. acaule, and S. tuberosum then backcrossed with 4x S. tuberosum continuously, and finally yielded 4x frost-tolerant interspecific accessions in 2005. In a word, the eight interspecific accessions in the present study was selected from advanced families of commersonii-acaule-tuberosum for cold hardiness and field tuberization (https://npgsweb.ars-grin.gov/gringlobal/cooperator?id=53268) (line 128-133).
- Chen, H.H.; Li, P.H. Characteristics of cold acclimation and deacclimation in tuber-bearing Solanum Plant Physiol. 1980, 65, 1146-1148.
- Bamberg, J.; Hanneman Jr, R.; Palta, J.; Harbage, J. Using disomic 4x(2EBN) potato species' germplasm via bridge species Solanum commersonii. Genome 1994, 37, 866-870.
- This study was carried out in 2018, how authors give validity of results?
Response: We express our gratitude for your constructive remarks. Actually, some of the F1 hybrids, particularly these genotypes in resistant/sensitive pools, were assessed their freezing tolerance by electrolyte leakage method (data not shown). The results of F1 hybrids frost/freezing tolerance determined by the natural frost method and electrolyte leakage method are relatively consistent.
We intended to conduct the third field frost test in 2019 and 2020 to further support our findings, unfortunately, the COVID-19 pandemic outbreak resulted in the abortion of field nature frost during the laboratory lockdown period. Not only that, COVID-19 delayed the overall experimental schedule, including BSA and SNP marker development.
- Many errors must be replaced with subscript/superscript with F (filial generation)
Response: We are very sorry for the writing errors caused by our oversight. Through consulting a large number of classic literature, we found that the subscript with F (filial generation) is accurate academic standards. Hence, we have corrected it throughout the revision as the reviewer's suggestion. Thanks again.
- This article must be rephrased with the help of native English speaker and it should not be accepted in current form for publication.
Response: We will be happy to edit the text further, based on helpful comments from the reviewers. We have carefully checked and improved the English writing in the revised manuscript accordingly. Besides, we have further polished the English writing by Elsevier's language service in the revised manuscript, and Editing Certificate was attached as the supplemental material. We tried our best to improve the manuscript and we hope the revised paper will be clearer and more accurate on expressions.
Reviewer 3 Report
In the manuscript, “Molecular marker-assisted selection for frost tolerance in a diallel population of potato” Tu et al. reported the potential frost-resistant SNP markers in three candidate regions related to frost tolerance, that were identified from progenies of frost-tolerant advanced breeding lines and eight cultivars. These genetic loci for frost tolerance were identified by molecular marker-assisted selection (MAS) system bulked segregant analysis (BSA). Their findings are of potential application in frost-tolerant germplasm improvement. They provided a large amount of data. and the title and abstract are appropriate for the content of the text.
As explained below, the major concern regarding the experimental data in form of the main figures and tables to be well-founded, figures and table should be reformed as suggested in the comments to make it easier for readers to interpret.
Comments
Line 47: Please give the full name of EBN.
Line 52: endosperm balance number (EBN) should be removed.
Line 70; Please rewrite the sentence “What is more interesting is that a frost-tolerant potato variety designated Alaska Frostless was selected in 1961 at Matanuska, which combines the frost tolerance of S. acaule with desirable characteristics of potato cultivars and can tolerate field frosting as low as -3 °C for 2 hours”
Line 73: “Alaska Frost less had significantly less total yield and slower vine growth than standard varieties, so even though its culinary characteristics are very good, its large-scale cultivation and application in potato production have been limited.” does seem right Grammarly.
Line 96: Please rewrite “To date, there has been no follow-up concentrating on genetic control in S. commersonii regarding frost (freezing) tolerance, and a molecular marker-assisted selection system for frost-tolerant potato breeding following natural frost under field conditions is still lacking.”
Line 249-252, the average score (AS) numbers can not find in table 1.
Line 255 the AS number cannot find in found in table 1 eighter.
Line 273, should be “A total of 406 progenies were …..”
Table 1 is not easy for the readers to interpret, please reformat it to make it self-explanatory by providing the time and geographic sites where experiments were performed to replace the environment I, II…., please provide the average scores that are mentioned in the text. Also please provide information the which are the interspecific hybrids plants.
Figure 1, please provide representative individual plant images to show the frost-tolerant and frost-sensitive phenotype. The images in Fig. are too blur to tell hybrids or parents plants performance to frost stress.
Line 313- 314, average frequency (4.13 ± 1.89 vs. 4.00 ± 2.20), same problem as previously mentioned, these numbers can not find in table 2. Please check them throughout the manuscript.
Fig.3 Please explain why only analyze the environment I and IV.
Line 327-330, “These sequences were aligned to the reference genome DM_V6_1 using BWA/SAM software with an average alignment rate of 82.98% and 85.28% between RP and SP. The average genome coverage and depth were 75.96% and 23.47×, respectively.” Which data support these numbers? Please add the reference data.
Figure 4 Please provide the number of the threshold in the figure legend.
Line 372, why the Except for is highlighted?
Line 374, where could the readers find the “was 66.56%” in table 3?
Author Response
Author's Reply to the Review Report (Reviewer 3)
Comments and Suggestions for Authors:
In the manuscript, “Molecular marker-assisted selection for frost tolerance in a diallel population of potato” Tu et al. reported the potential frost-resistant SNP markers in three candidate regions related to frost tolerance, that were identified from progenies of frost-tolerant advanced breeding lines and eight cultivars. These genetic loci for frost tolerance were identified by molecular marker-assisted selection (MAS) system bulked segregant analysis (BSA). Their findings are of potential application in frost-tolerant germplasm improvement. They provided a large amount of data. and the title and abstract are appropriate for the content of the text.
As explained below, the major concern regarding the experimental data in form of the main figures and tables to be well-founded, figures and table should be reformed as suggested in the comments to make it easier for readers to interpret.
Response: Thank you very much for the positive comments and constructive suggestions. Please find the following detailed responses to your comments and suggestions.
Comments
- Line 47: Please give the full name of EBN.
Response: Thank you for your valuable comments. We have corrected it in lines 48 as the reviewer's suggestion.
- Line 52: endosperm balance number (EBN) should be removed.
Response: Thank you for your constructive suggestion. We have deleted the phrase and revised the structure of the sentence in order to make it more readable as the reviewer's suggestion (Line 54 to 57).
- Line 70; Please rewrite the sentence “What is more interesting is that a frost-tolerant potato variety designated Alaska Frostless was selected in 1961 at Matanuska, which combines the frost tolerance of S. acaule with desirable characteristics of potato cultivars and can tolerate field frosting as low as -3 °C for 2 hours”
Response: We apologize for the confusion caused by our previous description. Since the sentence structure is not clear, we have rewritten the sentence as “What is more interesting is that a frost-tolerant potato cultivar called Alaska Frostless, which combines the frost tolerance of S. acaule with desirable characteristics of potato cultivars and can tolerate field frosting as low as -3 °C for 2 hours, was selected in 1961 at Matanuska” (Line 78 to 82).
- Line 73: “Alaska Frost less had significantly less total yield and slower vine growth than standard varieties, so even though its culinary characteristics are very good, its large-scale cultivation and application in potato production have been limited.” does seem right Grammarly.
Response: We are very sorry for the unclear statement. This sentence does seem grammatically incorrect and poor expression, so we have modified it as “Alaska Frostless had significantly lower total yield and slower vine growth than common potato cultivars, resulting in the large-scale cultivation and application being limited despite its culinary characteristics being impeccable” (Line 82 to 85).
- Line 96: Please rewrite “To date, there has been no follow-up concentrating on genetic control in S. commersonii regarding frost (freezing) tolerance, and a molecular marker-assisted selection system for frost-tolerant potato breeding following natural frost under field conditions is still lacking.”
Response: We are very sorry for the ambiguous expression. Hence, we corrected it as “To date, there has been no follow-up research on S. commersonii genetic control in frost (freezing) tolerance, and a molecular marker-assisted selection system for frost-tolerant potato breeding is still blank” (Line 110 to 113).
- Line 249-252, the average score (AS) numbers can not find in table 1.
Response: We apologize for the lack of clarity concerning the average score (AS) data of the previous manuscript. The AS data here refers to the average frost resistance of 8 cultivated potatoes or 8 interspecific hybrids in the corresponding environment. For example, the AS data 3.49 ± 0.68 is the average frost tolerance of eight cultivated potatoes in Environment I, in which frost tolerance is 3.60±0.55, 2.75±0.50, 3.20±1.30, 2.75±0.50, 3.60±0.55, 3.25±0.50, 4.00±0.00, and 4.80±0.45, respectively. Since it was not clearly stated in the previous article, we have rewritten the sectional content in revision (lines 271-293).
Besides, these data have been viewed in Fig 2a,b, which conducted the comparative analysis for frost tolerance between cultivated potatoes and interspecific hybrids in various environments. Given that the data presented in Table 1 are original data from cultivated potatoes and interspecific hybrids, and it is not easy for the readers to interpret accordingly. There is absolutely no need to put it in the revised manuscript after careful consideration based on the constructive comments of reviewers. Therefore, we would like to adjust Table 1 to Supplement Table S3 in the Supplement data file.
- Line 255 the AS number cannot find in found in table 1 eighter.
Response: We are deeply appreciative of the correction to our prior imprecise statement. As stated above (question 6), the AS data here refers to the average frost resistance of 8 cultivated potatoes or 8 interspecific potato hybrids in the corresponding environment. We have corrected it carefully in the revised manuscript (line 271-293).
- Line 273, should be “A total of 406 progenies were …..”
Response: We are very sorry for the writing errors caused by our oversight. We have corrected it in the revised manuscript according to the review's suggestion (line 308).
- Table 1 is not easy for the readers to interpret, please reformat it to make it self-explanatory by providing the time and geographic sites where experiments were performed to replace the environment I, II…., please provide the average scores that are mentioned in the text. Also please provide information the which are the interspecific hybrids plants.
Response: We deeply apologize for our loose data analysis, poor visualization, and ambiguous description concerning Table 1. As mentioned above (question 6,7), we will intend to adjust Table 1 to Supplement Table S3 in the Supplement data file due to its content being similar to Figure 2a,b. We have reformated it to Table S3 in the Supplement data file to make it self-explanatory by providing the time of frost trial, geographic sites, the information on interspecific hybrids, and the average of AS data. Besides, we have corrected some errors and reorganized this part of the content in revision (line 271-293), since it was not clearly stated in the previous manuscript.
- Figure 1, please provide representative individual plant images to show the frost-tolerant and frost-sensitive phenotype. The images in Fig. are too blur to tell hybrids or parents plants performance to frost stress.
Response: Thanks for your kind suggestions, which are valuable for improving the accuracy of the manuscript. We apologize that the image quality in our original Figure 1 is not very good, especially in Figure 1c,d. Therefore, we redesigned Figure 1 by adding some representative individual plant images and replacing them with higher-quality pictures. We believe that modifying Figure 1 (line 297), as the reviewer suggested, would be clearer to review in the revised version.
- Line 313- 314, average frequency (4.13 ± 1.89 vs. 4.00 ± 2.20), same problem as previously mentioned, these numbers can not find in table 2. Please check them throughout the manuscript.
Response: We apologize for any ambiguous description in Table 2 and we appreciate your attention to detail. The average frequency of the frost-tolerant parents was calculated by the average of 8 interspecific hybrids between RP (resistant pool) and SP (sensitive pool), which have been updated by adding a column of average data in Table 1 (line 365).
- 3 Please explain why only analyze the environment I and IV.
Response: In the present study, two field frost trials concerning the cross parents firstly were performed under low temperature based on the local weather forecast station at Wuhan (Environments I, II, III, IV) and Luoyang (Environments V, VI, VII) (Fig S1). On the basis of the frost tolerance of multiple parents under different field experiments, the frost test in Luoyang either directly froze all the cultivated potatoes to death or hardly any damage was done (Fig S1; Fig 2a,b; Table S3), which is not suitable for dif-ferentiating various grades of resistant breeding materials. In contrast, the appropriate frost time and temperature in Wuhan enabled the effective characterization of various degrees of frost tolerance for potato germplasm materials (Fig S1, Table S3). Given the lowest temperature recorded by recorder along with the frost performance of multiple parents across the seven environments (Fig S1, Table S3), Environments I and IV seemed to effectively characterize various degrees of frost tolerance in the F1 hybrids (line 310-320).
- Line 327-330, “These sequences were aligned to the reference genome DM_V6_1 using BWA/SAM software with an average alignment rate of 82.98% and 85.28% between RP and SP. The average genome coverage and depth were 75.96% and 23.47×, respectively.” Which data support these numbers? Please add the reference data.
Response: We are grateful for your attentive observation and for bringing this matter to our attention. These reference data are presented in Table S5. We apologize for any inconvenience caused by our oversight and we have added the reference data in the text (line 375). In addition, I would like to briefly explain that since there is only Unmapped data in previous Table S4, and the Mapped data is 100% minus the Unmapped value. Hence, for better presentation, we added a row with Mapped data below the Unmapped data in the revised Table S5.
- Figure 4 Please provide the number of the threshold in the figure legend.
Response: We are sorry that we did not describe clearly the number of the threshold in the figure legend. We sincerely appreciate the reviewer for bringing up the issue. Therefore, we have already supplemented the crucial data of the threshold and reorganized the content of the figure legend (line 397).
- Line 372, why the Except for is highlighted?
Response: We apologize for the issue of the format in this highlighted section, which may result from the hyperlink incorrectly inserted when the previous manuscript converted .doc format to .pdf format. We appreciate your attention to detail and we have removed this hyperlink in the revised manuscript.
- Line 374, where could the readers find the “was 66.56%” in table 3?
Response: Thank you for taking the time to review our manuscript and pointing out the confusion with the number in the text. Firstly, the data 66.56% was the average goodness of fit of nine SNP markers correlated with frost tolerance. The goodness of fit was calculated by the number of SNP genotype '1' in the resistant pool and SNP genotype '0' in the sensitive pool divided by the total SNP genotypes for each SNP marker. However, due to our carelessness calculations, the average goodness of fit of nine SNP markers should be 73.89% instead of 66.56% (line 426). We sincerely apologize for the data calculation errors caused by our carelessness.
Round 2
Reviewer 1 Report
Manuscript is improved significantly so, can be accepted for publication in the journal.
Author Response
We gratefully thank you for the precious time the reviewer spent making constructive remarks in this previous manuscript. It is a great honor to have your recognition for this work.

Reviewer 2 Report
The certificate of English editing is not submitted with revised version.
One response of my query is still stands that how authors have fulfilled the assumptions of diallel cross in their experiments? I'm not satisfied from their response if this crop. Authors must the articles published by Hayman and Jinks.
Author Response
- The certificate of English editing is not submitted with revised version.
Response: We apologize for the lack of the certificate of English editing when we submitted the last revised version. We have submitted the certificate of English editing as a supplement this time.
- One response of my query is still stands that how authors have fulfilled the assumptions of diallel cross in their experiments? I'm not satisfied from their response if this crop. Authors must the articles published by Hayman and Jinks.
Response: We gratefully thank the reviewer for reading our paper carefully and pointing out this issue. Jinks and Hayman's method of analysis of diallel crosses (Hayman 1954) have been used to study genetic variation relationships for quantitative traits. Their method should meet the following assumptions: (a) diploid segregation, (b) no difference between reciprocal crosses, (c) no non-allelic interaction, (d) no multiple allelism, (e) homozygous parents, and (f) genes independently distributed across the parents. Due to strict assumptions, this methodology criticized the validity of its assumptions and analyzed the consequences of their failure (Baker 1978).
The cultivated potato, a vegetatively propagated clone, has tetrasomic inheritance and the genetic structure of the populations used as parents by potato breeders is complex; conventional biometrical-genetic techniques are therefore inapplicable to the crop. The tetraploid status of S. tuberosum combined with the fact that potential parents are almost invariably highly heterozygous complicates the basic assumptions of many of the biometrical techniques and, hence, makes such approaches less powerful in practical breeding than with diploid inbreeding species. However, the use of combining abilities (Griffing 1956), which are statistical parameters, independent of the genetic status of the crop, offers an alternative approach of considerable potential for the breeder.
Although experimental designs such as diallel crosses (Griffing's I method) are used infrequently by potato geneticists because it is difficult to obtain a complete series of crosses between a group of genotypes as required by the designs, some half diallel cross or partial diallel crosses have been conducted as Griffing's diallel mating designs (Griffing's II, III, IV method) to perform genetic analysis of combining ability. Killick (1977) performed genetic variation for nine traits of importance in tetraploid potatoes by a half-diallel set of crosses with six parents according to Griffing's IV method. Hassanpanah et al (2016) conducted a hybridization of potato cultivars with a half diallel cross using Griffing's III method and selected 397045-13, 397031-16, and 397067-6 as promising clones. Four potato cultivars were used as parents and crossed by mutual hybridization based on a half diallel cross using Griffing's III method (Darabad et al 2020). In the present study, the crossing experiments of the 16 parents were designed for half diallel crossing as described by Griffing’s II method, which includes all possible crosses between the parents involved in the cross in one direction. Finally, only 23 cross combinations yielded viable seeds because of unilateral incompatibility or self-incompatibility.
- Hayman B. The theory and analysis of diallel crosses. Genetics, 1954, 39: 789.
- Baker R. Issues in diallel analysis. Crop Sci, 1978, 18: 533-536.
- Griffing B. Concept of general and specific combining ability in relation to diallel crossing systems. Australian journal of biological sciences, 1956, 9: 463-493.
- Killick RJ. Genetic analysis of several traits in potatoes by means of a diallel cross. Ann. Appl. Biol, 1977, 86: 279-289.
- Darabad GR, Hassandokht MR, Hassanpanah D, Mousavi A. Diallel cross in potato cultivars (Solanum tuberosum L.) and evaluation of their progenies under deficit water stress. Acta Agrobotanica, 2020, 73: 1-9.
- Hassanpanah D, Hassanabadi H, Hoseinzadeh AA, Soheli B, Mohammadi R.Factor analysis, AMMI stability value (ASV) parameter and GGE BI-plot graphical method ofquantitative and qualitative traits in potato genotypes. Journal of Crop Ecophysiology, 2016, 10: 731–748.

Reviewer 3 Report
In the submitted review manuscript Tu et al., describes the advantages of using diallel populations of potatoes to broaden and improve frost-tolerant potato germplasm. This is an exciting and well-summarized review paper. This revised draft addressed all my previous concerns. Professionals edited the whole draft. New data was added to cover the major points this manuscript focuses on. The title and abstract are appropriate for the content of the text. I would like to suggest editor accept this paper.
1: Please add scale bars in figure 1.
2: Please use the same fonts for all figures and tables.
Author Response
1: Please add scale bars in figure 1.
Response: We are very sorry for our negligence with the scale bars in Figure 1. Considering the Reviewer’s suggestion, we have added scale bars in each sub-figure in Figure 1.
2: Please use the same fonts for all figures and tables.
Response: We would like to thank you for your careful reading, helpful comments, and constructive suggestions, which have significantly improved the presentation of our manuscript. As the reviewer suggested that we have unified the same font (Palatino Linotype) for all figures and tables, consistent with the font in the manuscript.

Round 3
Reviewer 2 Report
Significant changes are made in the article, but still there is room for improvement in Introduction, presentation of results and discussion